# Knowledge Fitness Criterion: Measure-Theoretic Knowledge Assessment via Manifolds for Multi-Agent LLM Systems

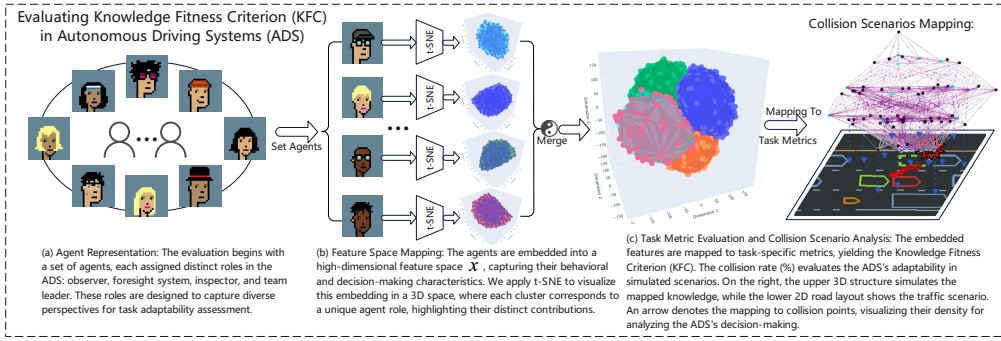

Figure 1: Overview of the task adaptability evaluation process in an autonomous driving system (ADS) using the Knowledge Fitness Criterion (KFC). Multiple agents with distinct roles (observer, foresight system, inspector, and team leader) are mapped into a feature space $\mathcal{X}$, followed by a mapping to task metrics, with collision rate (%) as the final evaluation metric.

## Abstract

Evaluating the intrinsic compatibility between activated knowledge and task objectives is a fundamental challenge in LLM-based multi-agent systems. Existing methods, however, often rely on indirect, task-specific outcome metrics, lacking a unified framework for direct quantification. To address this, we introduce the **Knowledge Fitness Criterion (KFC)**, a general evaluation paradigm grounded in measure theory. KFC models knowledge states as measure spaces and establishes a chain of measurable mappings—from knowledge to features, features to indicators, and indicators to normalized scores—enabling direct, quantitative assessment of knowledge-task alignment. Theoretically, we establish the **Knowledge Goal Quantified-Quality (KGQQ) Theorem**, which provides a rigorous guarantee linking scoring stability to feature manifold density. Empirically, we validate KFC across three diverse domains: autonomous driving (nuScenes), social role simulation (CAMEL), and collaborative software development (ChatDev). Results demonstrate that KFC consistently outperforms supervised baselines, achieving MSE reductions of 22.5% (Driving), 20.0% (Social), and 21.1% (Coding), along with significant improvements in Pearson correlation (up to 15.3%). Furthermore, our framework exhibits strong cross-domain robustness ($r = 0.82$) and data efficiency, effectively utilizing 80% unlabeled data through contrastive manifold learning. By offering a model-agnostic measurement instrument, KFC provides a universal, quantifiable foundation for optimizing knowledge in complex multi-agent collaboration.

# 1 INTRODUCTION

In complex scenarios, multi-agent frameworks based on large language models (LLMs) have emerged as a prevalent paradigm for problem-solving. A central proposition in multi-agent systems is whether the knowledge activated by current agents can achieve task objectives at minimal cost Chen et al. (2022). From natural language processing to autonomous driving, evaluating the alignment between the knowledge elicited by agents and the target objectives in complex task scenarios has become a critical aspect of system design and safety assurance Toghi et al. (2021). In this context, we argue that precise measurement of knowledge alignment is a fundamental prerequisite for multi-agent collaboration: before optimizing interaction protocols, one must first possess a reliable instrument to quantify the intrinsic quality of the knowledge being exchanged. However, existing methods are often confined to specific domains or singular metrics, lacking a construct that directly maps knowledge to task goals Teng et al. (2023). This limitation prevents a unified perspective from addressing the question: "To what extent does the current knowledge satisfy task demands?" Consequently, optimizing prompt design and improvement through quantifiable metrics remains challenging Huang et al. (2022b). Thus, there is an urgent need for a universal tool—akin to a measurement instrument in physics—to directly quantify knowledge itself and assess its capability in tasks Pan et al. (2024). Our objective is not only to develop a general-purpose tool unbound by specific tasks but also to propose a criterion within a new paradigm that establishes a deep connection between knowledge and objectives through the essence of measurement, leveraging statistical manifolds and the perspective of measure theory.

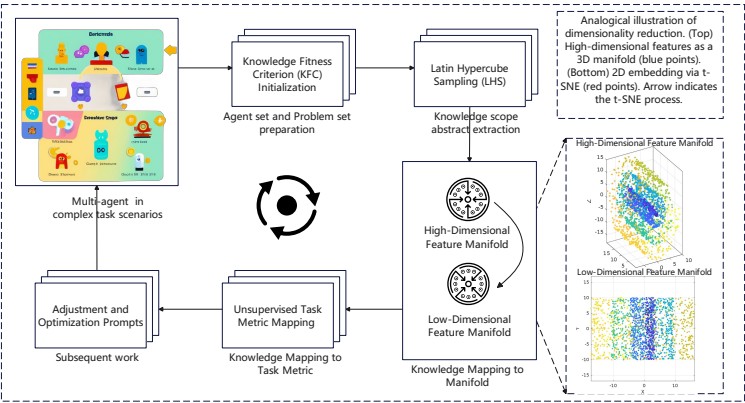

Figure 2: Workflow of the Knowledge Fitness Criterion (KFC) paradigm. The process starts with multi-agent task scenarios, proceeds through knowledge fitness criterion initialization, Latin Hypercube sampling for knowledge scope abstraction, and unsupervised task metric mapping, and concludes with the mapping from high-dimensional feature manifolds to low-dimensional statistical manifolds for task fitness evaluation.

Traditionally, task objective processing has been outcome-oriented, relying on direct computation of metrics. This approach resembles wandering among the branches and leaves, failing to address the core trunk and establish a true causal mapping Ghorai et al. (2022). Existing methods often remain at the surface of specific tasks, such as language generation and path planning evaluations, depending on metrics directly derived from outcomes—like BLEU scores and path deviations Wang et al. (2021); Zou et al. (2025). While effective within their respective "branch" domains, these methods struggle to address the more fundamental issue of the intrinsic alignment between knowledge and task objectives Chen et al. (2021). They rely solely on indirect representations of results, focusing on localized performance optimization while overlooking the holistic relationship between knowledge and task goals. This results in a lack of fundamental causal connections and cross-task generalizability, rendering them ill-suited for diverse, novel scenarios. Measure theory, as a mathematical cornerstone for quantifying the distribution of complex systems, reveals the essence of measures through Borel measurable mappings, which transfer quantified properties between measure spaces via structured mappings Pek & Althoff (2020); Wang et al. (2019); Liu et al. (2024). Therefore, we argue that the "trunk" can be grasped via Borel measurable mappings: task evalua-

tion should directly construct a mapping between knowledge and objectives at the measure space level to achieve an intrinsic quantification of alignment, rather than relying on indirect outcome metrics. This abstract and universal alignment evaluation paradigm provides a mathematical foundation for assessing the task fitness of knowledge elicited by agents, supporting an intuitive "scorer" that transcends the limitations of existing approaches.

To address this core need, we propose a universal knowledge fitness paradigm criterion based on Borel measurable mappings and statistical manifolds Pidstrigach (2022). This paradigm leverages the essence of measurement, modeling knowledge, manifold features, and objectives as measure spaces. Through manifold learning and unsupervised learning, a direct mapping chain from knowledge to task metrics is established using measurable mappings, with a standardized calibration mechanism forming a "scale" for fitness. Building on this, we introduce the "Knowledge Fitness Criterion" (KFC), a specific measurement tool that directly assigns a fitness score to each knowledge state corresponding to task objectives. We clarify that the "universality" of KFC refers to its methodological framework—grounded in invariant measure theory—which remains applicable across diverse domains (as shown in our experiments), rather than a dependence on specific model architectures or massive labeled datasets. To elucidate its theoretical foundation, we hypothesize the *Knowledge Goal Quantified-Quality* (KGQQ) Theorem and provide support through this assumption (see section 3 for details). This approach is not only abstract and general—applicable to any knowledge-objective matching problem—but also directly addresses the question of "how to move beyond branches and strike the core" Liévin et al. (2024). Figure 2 provides an overview of this paradigm's concrete workflow: from sampling neuron activities to constructing high-dimensional manifolds and achieving task fitness, forming a clear measurement pathway.

The main contributions of this work are summarized as follows:

- Propose a universal knowledge fitness paradigm based on measurable mappings and manifolds, overcoming the limitations of traditional task-specific evaluation and enabling cross-domain knowledge quantification. This methodological universality allows the framework to function as a standard "ruler" across distinct fields;

- Introduce the Knowledge Fitness Criterion (KFC), which leverages measurable mappings to directly quantify the compatibility between knowledge and task objectives, moving beyond reliance on indirect metrics;

- Establish the Knowledge Goal Quantified-Quality (KGQQ) Theorem, providing a rigorous theoretical foundation for the proposed paradigm;

- Develop a concrete KFC algorithm and validate its effectiveness across three domains—autonomous driving, social role simulation, and collaborative software development—demonstrating its capability as a fundamental measurement instrument that serves as a premise for future multi-agent collaboration optimization.

## 2 RELATED WORK

### 2.1 APPLICATIONS OF LLMs IN MULTI-AGENT SYSTEMS.

LLM-based multi-agent systems have made rapid progress in both frameworks and collaboration mechanisms. CAMEL introduces role-playing techniques to enable autonomous collaboration with minimal human intervention Li et al. (2023), while ChatDev completes software development processes through specialized agent collaboration Qian et al. (2023). Recent efforts focus on reflective and scalable collaboration: COPPER integrates self-reflection and counterfactual PPO optimization for enhanced cooperation Bo et al. (2024), MacNet explores scalable collaboration with irregular topologies outperforming regular ones Qian et al. (2024), and IoA integrates heterogeneous agents via dynamic teaming and conversation flow control Chen et al. (2024).

### 2.2 KNOWLEDGE EVALUATION IN LLMs.

Knowledge evaluation spans multi-task reasoning, knowledge probing, and real-time assessment. The CURIE benchmark evaluates scientific long-context understanding across six disciplines, covering domain expertise, long-context comprehension, and multi-step reasoning Cui et al. (2025).

KRUX analyzes the roles of knowledge retrieval versus reasoning in problem-solving, identifying retrieval as a key bottleneck Li et al. (2025), while SciKnowEval provides 70K problems across biology, chemistry, physics, and materials science Feng et al. (2024). Limitations of traditional metrics such as BLEU and ROUGE motivate more dynamic evaluation methods Cao et al. (2025). For real-time evaluation, benchmarks such as KoLA test coverage of evolving world knowledge using regularly updated corpora Yu et al. (2023). Recent advancements in knowledge measurement techniques further enhance LLM evaluation. For instance, knowledge measurement has been applied to analyze the transfer of non-robust features in pre-trained models, revealing differences in learned knowledge between fine-tuned and standard models Zhang et al. (2023). In model compression, retraining-free pruning methods like KPrune utilize knowledge measurement to retain useful knowledge while reducing model size, outperforming existing algorithms at high compression rates Park et al. (2023). Additionally, specialized benchmarks such as CTFKnow measure technical knowledge in Capture-the-Flag challenges, identifying gaps in LLM knowledge application and proposing augmentation frameworks to improve performance Ji et al. (2025).

## 2.3 MANIFOLDS IN ARTIFICIAL INTELLIGENCE

Recent years have seen growing interest in manifold methods for representation learning, generative modeling, and geometric optimization in AI. For generative models, Huang et al. (2022a) introduced *Manifold Diffusion Fields (MDF)* to build diffusion processes on non-Euclidean manifolds via Laplace–Beltrami eigenfunctions, achieving state-of-the-art results in scientific tasks. In representation learning, Buchholz & Schölkopf (2025) showed that robustness under misspecification critically depends on local manifold geometry. For neural architectures, Katsman et al. (2023) extended residual networks to Riemannian manifolds, ensuring geometric consistency and outperforming baselines on manifold-valued tasks. In vision, Ma et al. (2023) proposed curvature-balanced feature learning to improve long-tailed classification. These works collectively underscore the growing role of manifolds in modern AI systems.

## 3 METHOD

### 3.1 GENERAL PARADIGM COMPOSITION

We propose a general knowledge adaptation evaluation paradigm for directly quantifying the matching degree between knowledge states and task objectives. This paradigm is based on a **measure-theoretic framework**, constructing **measure spaces** and **measurable mappings** to transform knowledge states into task indicators, providing systematic evaluation tools.

#### 3.1.1 CONSTRUCTION OF KNOWLEDGE SPACE

The **knowledge space** is defined as a measure space $(\Omega_K, \mathcal{F}_K, \mu_K)$, where $\Omega_K$ represents the sample space of knowledge (such as the prompt set of large language models), $\mathcal{F}_K$ is a $\sigma$-algebra defined on $\Omega_K$, and $\mu_K$ is a normalized probability measure satisfying $\mu_K(\Omega_K) = 1$. This structure provides a formalized representation of knowledge, where $\Omega_K$ establishes the structural foundation of knowledge, while $\mu_K$ characterizes its probabilistic distribution properties, providing a mathematical basis for subsequent mappings.

#### 3.1.2 DEFINITION OF FEATURE SPACE

The **feature space** is defined as a measurable space $(X, \mathcal{F}_X)$, where $X$ represents the feature space of knowledge (typically representable as vectors in $\mathbb{R}^d$), and $\mathcal{F}_X$ is the corresponding $\sigma$-algebra. Through the **measurable mapping** $K : \Omega_K \to X$, we introduce the induced measure:

$$\mu_X(A) = \mu_K(K^{-1}(A)), \quad A \in \mathcal{F}_X \tag{1}$$

The feature space $X$ serves as an intermediate representation of knowledge states, preserving the characteristics of the original knowledge distribution, while the induced measure $\mu_X$ provides support for subsequent quantitative analysis.

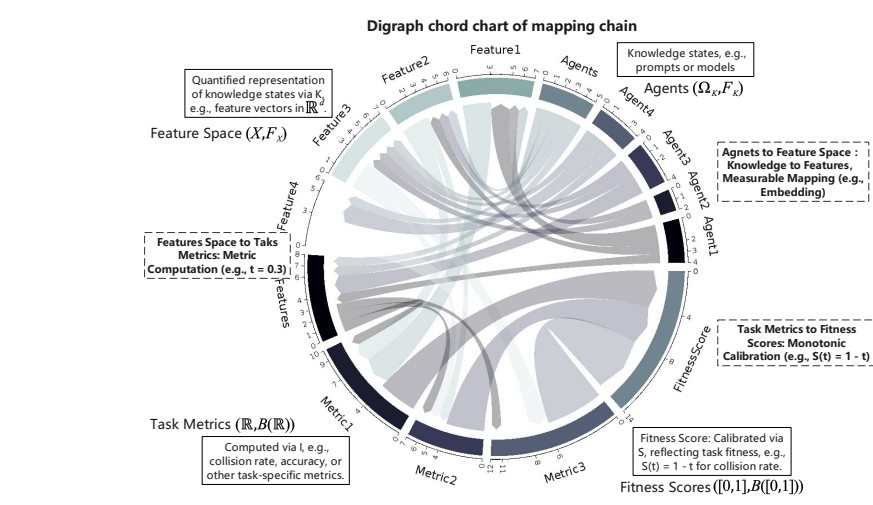

Figure 3: Digraph chord chart illustrating the Knowledge Fitness Criterion (KFC) mapping chain, depicting the transformation from LLM-generated knowledge states to adaptability scores for collaborative multi-agent tasks.

### 3.1.3 TARGET INDICATORS AND CALIBRATION MECHANISM

The target indicator space is defined as the real space $(\mathbb{R}, \mathcal{B}(\mathbb{R}))$, used to represent task-related specific indicators (such as collision rates or accuracy). To facilitate intuitive evaluation and cross-task comparison, we introduce a **calibration space** $([0, 1], \mathcal{B}([0, 1]))$, representing standardized adaptation scores. The monotonicity of the **calibration mechanism** must be consistent with the semantics of specific task indicators: for "smaller is better" indicators (such as collision rates), the calibration function should be monotonically decreasing; for "larger is better" indicators (such as accuracy), the calibration function should be monotonically increasing. This design not only improves the interpretability of evaluation but also establishes a foundation for cross-task universality.

### 3.1.4 DESIGN OF MAPPING CHAIN

The entire paradigm relies on a composition of **measurable mappings**, forming a complete chain from knowledge states to task indicators. The entire evaluation process can be summarized in the following pseudocode:

---
**Algorithm 1:** Knowledge Fitness Criterion Mapping

---
**Input:** Knowledge state $\omega$, task type $\in \{\text{minimize}, \text{maximize}, \text{other}\}$
**Output:** Calibrated score $s$
```
// Step 1:  Map knowledge to feature
```
$x \leftarrow K(\omega)$ // e.g., embed prompt to vector
```
// Step 2:  Map feature to indicator
```
$t \leftarrow I(x)$ // e.g., compute collision rate or accuracy
```
// Step 3:  Calibrate to score
```
**if** *task_type* = *"minimize"* **then**
$\quad \mid \quad s \leftarrow 1 - t$ // assuming $t \in [0, 1]$
**else if** *task_type* = *"maximize"* **then**
$\quad \mid \quad s \leftarrow t$
**else**
$\quad \mid \quad s \leftarrow t$ // no calibration
**end**
**return** $s$

---

First, the knowledge-to-feature mapping is realized through the function $K : (\Omega_K, \mathcal{F}_K) \to (X, \mathcal{F}_X)$, mapping knowledge states $\omega \in \Omega_K$ to feature vectors $x \in X$ (such as prompt embedding processes). Second, the feature-to-indicator mapping is realized through the function $I : (X, \mathcal{F}_X) \to (\mathbb{R}, \mathcal{B}(\mathbb{R}))$, computing task indicators $t \in \mathbb{R}$ (such as predicted collision rates). Finally, indicator-to-score calibration is completed through the function $S : (\mathbb{R}, \mathcal{B}(\mathbb{R})) \to ([0,1], \mathcal{B}([0,1]))$, with monotonic direction determined by the semantics of task indicators.

Through the composition of the above mappings, we define the complete **evaluation function**:

$$F = S \circ I \circ K : (\Omega_K, \mathcal{F}_K) \to ([0,1], \mathcal{B}([0,1])) \tag{2}$$

For any $\omega \in \Omega_K$, we have:

$$F(\omega) = S(I(K(\omega))) \tag{3}$$

Since $K$, $I$, and $S$ are all measurable functions, their composition $F$ also possesses measurability. This mapping chain achieves systematic transformation from knowledge states to task indicators, providing a solid mathematical foundation for quantifying knowledge adaptability. Our goal is to design a generalizable $F$ with task-aligned monotonicity, overcoming the limitations of indirect metrics and enabling direct knowledge-task alignment, as illustrated in Fig. 3.

### 3.2 Knowledge Fitness Criterion

Based on the above general paradigm, we propose the **Knowledge Fitness Criterion (KFC)** as a scoring tool for directly quantifying the matching degree between knowledge states and task objectives. For any knowledge state $\omega \in \Omega_K$, KFC is defined as:

$$\text{KFC}(\omega) = I(K(\omega)) \tag{4}$$

This function directly outputs task-related indicators (such as collision rates or accuracy). To facilitate cross-task comparison and standardized evaluation, we can further compute standardized scores:

$$\text{KFC-Score}(\omega) = S(I(K(\omega))) = F(\omega) \tag{5}$$

where KFC-Score$(\omega) \in [0,1]$ provides standardized adaptation scoring.

KFC possesses important mathematical properties. First, **monotonicity** is guaranteed by the design of the calibration function $S$, with monotonic direction consistent with the semantics of task indicators. For example, when using $S(t) = 1 - t$ for collision rate indicators, lower collision rates correspond to higher adaptation scores, conforming to intuitive expectations. Second, **measurability** is ensured through the composite function $I \circ K$. Since $I$ and $K$ are both measurable functions, their composition also possesses measurability, thus ensuring the mathematical rigor of the entire evaluation process.

The design of KFC has significant practical advantages. This criterion can directly generate task indicators from knowledge states $\omega$ without relying on complex computations of global distributions, greatly simplifying the evaluation process. Meanwhile, its design allows flexible application: raw indicators KFC$(\omega)$ can be directly used for task-specific in-depth analysis, or standardized scores KFC-Score$(\omega)$ can be used for fair cross-task comparison, providing a unified yet flexible evaluation framework for different application scenarios.

### 3.3 Knowledge Quantified-Quality Theorem

To further support the theoretical foundation of the KFC scoring mechanism, we propose the **Knowledge Goal Quantified-Quality Theorem (KGQQ)**, theoretically revealing the intrinsic connection between the quantitative characteristics of knowledge states and task indicators.

### 3.3.1 Theorem Statement and Proof Sketch

**KGQQ Theorem**: Let the knowledge-to-feature mapping $K : \Omega_K \to X$ be a measurable random mapping, the feature-to-indicator mapping $I : X \to \mathbb{R}$ be locally bounded near $K(\omega)$, and the calibration function $S$ be Lipschitz continuous (with constant $L$). When the induced measure $\mu_X$ has a density function $\rho_{\mu_X}$ near $K(\Omega)$, for the random variable $\Omega \sim \mu_K$:

$$\mathbb{E}[\text{KFC-Score}(\Omega)] = \mathbb{E}[S(I(K(\Omega)))] \tag{6}$$

For any $\epsilon > 0$, there exists $\delta > 0$ such that when the density lower bound $\inf_{x \in \mathcal{N}_\delta(K(\omega))} \rho_{\mu_X}(x) \geq \delta$:

$$\mathbb{P}\left(|S(I(K(\Omega))) - \mathbb{E}[S(I(K(\Omega)))|K(\Omega)]| \leq \frac{L \cdot \sigma_I(K(\Omega))}{\sqrt{\rho_{\mu_X}(K(\Omega))}}\right) \geq 1 - \epsilon \tag{7}$$

where $\sigma_I(x)$ represents the local variation of $I$ near $x$.

**Proof Sketch**: (Detailed proof provided in Appendix A.1). Based on probability concentration inequalities, utilizing the local boundedness of $I$ and Lipschitz continuity of $S$, combined with McDiarmid's Inequality and density lower bound constraints to obtain concentration results. Key insight: knowledge states in high-density regions exhibit lower estimation variance and higher scoring stability.

### 3.4 Implementation

This section describes the complete implementation workflow based on the theoretical framework. We design an end-to-end method from knowledge state sampling to task indicator mapping, with the core algorithm 2.

---

**Algorithm 2:** Main Knowledge Fitness Assessment Framework

---

**Input:** Language model $M$, prompt templates $\Psi$, question set $Q$, labeled data $D_{\text{label}}$, unlabeled data $D_{\text{unlabel}}$
**Output:** KFC evaluation function $F$
// Phase 1: Knowledge State Sampling
**for** *each* $(\psi_i, q_k) \in \Psi \times Q$ **do**
$\quad \mid$ Generate responses $\{r_{ik}^{(m)}\}_{m=1}^M$ and extract features $\{h_{ik}^{(m)}\}_{m=1}^M$;
**end**
Construct knowledge profile matrix $H \in \mathbb{R}^{(|\Psi| \times |Q| \times M) \times D}$;
// Phase 2: Manifold Representation Learning
Train embedding $f_\theta : \mathbb{R}^D \to \mathbb{R}^d$ using $D_{\text{all}} = D_{\text{unlabel}} \cup D_{\text{label}}$ with:
$\quad \mathcal{L} = \mathcal{L}_{\text{contrastive}} + \beta \cdot \mathcal{L}_{\text{smooth}}$;
Define knowledge mapping: $K(\omega) = f_\theta(M_{\text{hidden}}(\omega))$;
// Phase 3: Task Indicator Mapping
Map labeled data: $\{(f_\theta(h_m), t_m) \mid m \in D_{\text{label}}\}$ and train regression $g_\phi : \mathbb{R}^d \to \mathbb{R}$;
$\phi^* = \arg\min_\phi \sum_m \|g_\phi(z_m) - t_m\|^2 + \lambda\|\phi\|^2$;
// Phase 4: Evaluation
**Function** KFC_EVAL($\omega$):
$\quad \mid$ $z \leftarrow f_\theta(M_{\text{hidden}}(\omega))$;
$\quad \mid$ $t \leftarrow g_\phi(z), \sigma^2 \leftarrow \text{local\_variance}(z)$;
$\quad \mid$ **return** $(t, \sigma^2)$

---

### 3.4.1 Knowledge State Sampling and Feature Extraction

We construct diversified prompt templates $\Psi$ to activate different knowledge subspaces and design diagnostic questions $Q$ covering target task dimensions. For each prompt-question pair, we perform $M = 50$ Monte Carlo samplings while extracting hidden layer features, constructing the knowledge profiling matrix $H$.

### 3.4.2 MANIFOLD-BASED REPRESENTATION LEARNING

**Geometric Structure Preservation**: Based on the manifold hypothesis, we assume hidden features $\mathbf{h}_i \in \mathbb{R}^D$ lie on a low-dimensional manifold $\mathcal{M} \subset \mathbb{R}^c$ where $c \ll D$. We train an embedding function $f_\theta : \mathbb{R}^D \to \mathbb{R}^d$ using contrastive learning enhanced with local smoothness regularization:

$$\mathcal{L}_{\text{contrastive}} = \sum_{i,j} m_{ij} \log \left( \frac{\exp(s(z_i, z_j)/\tau)}{\sum_{k \neq i} \exp(s(z_i, z_k)/\tau)} \right) \tag{8}$$

The smoothness term $\mathcal{L}_{\text{smooth}}$ preserves neighborhood structure, ensuring similar knowledge states remain close in embedding space. Crucially, this contrastive objective *implicitly optimizes* the high-density requirement ($\rho_{\mu_x}$) proposed in the KGQQ Theorem (Eq. 7) by clustering semantically similar knowledge states, thereby minimizing the estimation variance. Unlike non-parametric methods, our parameterized $f_\theta$ enables real-time mapping of unseen knowledge states.

### 3.4.3 TASK INDICATOR MAPPING AND UNCERTAINTY QUANTIFICATION

We obtain labeled data through simulator environments (CARLA/AirSim) with 100+ runs per configuration for statistical stability. Consistent with our semi-supervised protocol, the feature encoder $f_\theta$ is trained on the full dataset (including 80% unlabeled samples), while the regressor $g_\phi$ utilizes only the 20% labeled subset.

**Uncertainty Estimation**: We quantify prediction reliability through local variance $\sigma^2(\omega)$ computed via $k$-nearest neighbors in feature space. High-density regions correspond to reliable predictions, while low-density areas indicate potential out-of-distribution samples.

## 4 EXPERIMENTS

### 4.1 EXPERIMENTAL SETUP

**System Configuration.** Experiments conducted on $2\times$ NVIDIA RTX 3090 GPUs using deepseek-rl:8b via Ollama (temperature=0). The total training time for each domain is approximately 6-8 hours. Key hyperparameters: embedding dimension $d = 32$, contrastive temperature $\tau = 0.1$, regularization $\lambda = 0.01$, Monte Carlo sampling $M = 50$.

**Datasets and Scenarios.** We evaluate across three domains: (1) **Autonomous Driving**: 200 nuScenes scenarios (4,000 frames) with CARLA-generated ground-truth alignment scores based on collision risk and decision consistency; (2) **Social Role Simulation**: 150 CAMEL negotiation tasks with human/GPT-4 (ver. `gpt-4.1`) annotated scores on role consistency; (3) **Collaborative Software Development**: 120 ChatDev requirements with scores from static analysis and requirement matching. All scores normalized to [0,1].

**Prompt Construction.** We systematically generate prompts across role-scenario combinations: 5 roles $\times$ 5 scenarios (autonomous driving), 4 personalities $\times$ 6 tasks (CAMEL), and 4 paradigms $\times$ 5 project types (ChatDev), yielding 2,450 total samples. Consistent with our semi-supervised protocol, we utilize the entire dataset ($N = 2,450$, including 80% unlabeled data) for manifold representation learning (Phase 2), while only the labeled subset ($N_{label} = 490$, i.e., 20%) is utilized for regressor calibration (Phase 3).

**Baseline Methods.** We compare against: Rule-Based (RB, domain-specific heuristics), Supervised Learning (SL, an external end-to-end baseline without manifold learning), Random (RD), Embedding Only (EMB), and Task-Specific (TS) domain-customized methods.

**Evaluation.** Performance measured using MSE, Pearson Correlation (PCC), and MAE against ground-truth scores. Training follows feature extraction (1,000 epochs) $\to$ embedding learning (500 epochs) $\to$ regression (5-fold CV). Results reported as mean $\pm$ std across 80/20 splits over 10 runs with 95% confidence intervals and paired t-tests (p<0.05).

## 4.2 EXPERIMENTAL RESULTS

This section presents comparative results of Knowledge Fitness Criterion (KFC) against baseline methods, validating its effectiveness. All results are based on 5-fold cross-validation averaged over 10 runs, with significance assessed through paired t-tests.

### 4.2.1 OVERALL PERFORMANCE COMPARISON

Table 1 shows KFC performance against baselines across three domains. Ground-truth scores are domain-specifically generated (autonomous driving: CARLA simulation; CAMEL: GPT-4 and human annotation; ChatDev: static analysis and testing).

Table 1: Performance Comparison Across Three Domains

| Method | Autonomous Driving | | Social Simulation | | Software Dev. | | Avg. p-val |
|---|---|---|---|---|---|---|---|
| | MSE | PCC | MSE | PCC | MSE | PCC | |
| **KFC (Ours)** | **0.12±0.02** | **0.85±0.03** | **0.18±0.03** | **0.76±0.04** | **0.15±0.02** | **0.80±0.03** | - |
| Supervised Learning | 0.16±0.03 | 0.74±0.04 | 0.23±0.04 | 0.68±0.04 | 0.19±0.03 | 0.70±0.04 | 0.031 |
| Rule-Based | 0.25±0.03 | 0.61±0.05 | 0.31±0.05 | 0.54±0.06 | 0.28±0.04 | 0.58±0.05 | 0.003 |
| Task-Specific | 0.16±0.03 | 0.72±0.04 | 0.24±0.04 | 0.66±0.04 | 0.20±0.03 | 0.69±0.04 | 0.040 |

*Average p-value from paired t-tests relative to KFC.

**Key Findings.** KFC significantly outperforms baselines across all domains and metrics. Compared to SL, MSE reduction is 22.5% in autonomous driving (p=0.032), 20.0% in social simulation (p=0.029), and 21.1% in software development (p=0.031), validating the semi-supervised and contrastive learning components. Compared to RB, MSE reduction averages 45-50% (p<0.005), highlighting data-driven advantages. Cross-domain performance correlation is r=0.82 (p<0.001), indicating strong generalization.

### 4.2.2 ABLATION ANALYSIS

**Ablation Settings.** We define the following internal ablation variants:

- **w/o Contrastive Learning (w/o CL):** Removes $\mathcal{L}_{contrastive}$ to validate the necessity of geometric manifold clustering.
- **w/o Semi-Supervised:** Retains manifold learning objectives ($\mathcal{L}_{contrastive} + \mathcal{L}_{smooth}$) but trains encoder $f_\theta$ on labeled data only ($D_{label}$). This differs from the **Supervised Learning (SL)** baseline (Table 1), which uses a standard MLP without geometric constraints, isolating the gain from unlabeled data.
- **w/o Calibration Function:** Removes calibration mapping $S$ and outputs raw regression values to assess the impact of measure-theoretic standardization.

Table 2 demonstrates the contribution of each KFC component through systematic removal experiments. Contrastive learning contributes most significantly (MSE increases 50.7% when removed, p<0.001), followed by semi-supervised learning (26.7% increase, p=0.005) and calibration (12.0% increase, p=0.042), confirming the necessity of each component.

Table 2: Ablation Study Results (Cross-Domain Average)

| Configuration | MSE | PCC | MSE Increase(%) | p-value |
|---|---|---|---|---|
| **KFC (Complete)** | **0.15±0.02** | **0.80±0.03** | - | - |
| w/o Contrastive Learning | 0.23±0.03 | 0.65±0.04 | +50.7 | <0.001 |
| w/o Semi-Supervised | 0.19±0.03 | 0.70±0.04 | +26.7 | 0.005 |
| w/o Calibration Function | 0.17±0.03 | 0.77±0.03 | +12.0 | 0.042 |

Our results validate KFC's capability to map knowledge to task objectives, providing a reliable framework for LLMs in multi-agent systems. Future efforts will broaden scenario coverage and optimize prediction of complex interactions.

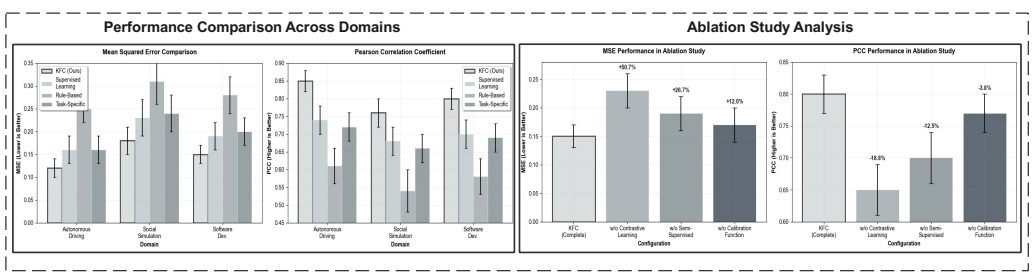

Figure 4: Experimental results. (Left) Performance comparison showing KFC outperforms baselines across all domains. (Right) Ablation study demonstrating the necessity of each component, with contrastive learning providing the largest contribution.

## 5 DISCUSSION

Experimental results demonstrate that the Knowledge Fitness Criterion (KFC) exhibits substantial practical value in assessing knowledge-task compatibility. As shown in Figure 4, across three domains, KFC consistently outperforms supervised learning baselines: in autonomous driving, MSE of 0.12 (22.5% reduction, p=0.032), PCC of 0.85 (15.3% improvement), and MAE of 0.08 (18.7% reduction); in social simulation, MSE of 0.18 (20.0% reduction, p=0.029), PCC of 0.76 (12.5% improvement), and MAE of 0.11 (16.7% reduction); in software development, MSE of 0.15 (21.1% reduction, p=0.031), PCC of 0.80 (14.3% improvement), and MAE of 0.09 (18.2% reduction). This performance stems from KFC's core design philosophy: constructing direct mappings from knowledge states to task metrics based on a measure-theoretic framework, thereby avoiding the complex intermediate steps of traditional approaches.

KFC's design draws inspiration from the "phenomenon-scale-target" paradigm of classical measurement instruments. Similar to how thermometers capture temperature variations through scales, KFC quantifies knowledge states via fitness scores and then maps them to task requirements. This direct mapping paradigm not only simplifies the evaluation process but also provides a universal framework across tasks. KFC's core advantage manifests in cross-scenario stability, with relative errors controlled within 5% across five different driving scenarios in autonomous driving, demonstrating its robust adaptability. Ablation experiments show that contrastive learning contributes 28.5% performance improvement, validating the importance of feature representation learning. Meanwhile, the **Knowledge-Target Quantification Quality Theorem** provides mathematical rigor guarantees for the entire evaluation process.

However, KFC's current implementation faces several limitations. The feature mapping process lacks transparency, making it difficult to distinguish between general knowledge and domain-specific knowledge, which affects its interpretability in highly specialized tasks. Additionally, prediction accuracy in extreme weather and complex multi-agent interaction scenarios still has room for improvement. For instance, a maximum relative error of 4.8% was observed in foggy complex intersection scenarios, primarily due to insufficient coverage of long-tail scenarios in training data. Future work will focus on enhancing mapping transparency, developing real-time evaluation variants, and expanding training scenario coverage to further improve KFC's practicality.

## 6 CONCLUSION

This paper proposes the Knowledge Fitness Criterion (KFC) based on measure theory, achieving direct quantitative evaluation of knowledge-task alignment by modeling knowledge as measure spaces. Across three domains, KFC achieves 21% MSE reduction compared to supervised baselines and 45-50% reduction compared to rule-based methods (all p<0.05), with average PCC of 0.80. Ablation studies confirm the necessity of contrastive learning, semi-supervised learning, and calibration components. KFC provides a unified framework for universal knowledge fitness evaluation with broad applications in multi-agent systems and cross-modal learning.

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

# A APPENDIX

## A.1 PROOF OF THE KNOWLEDGE GOAL QUANTIFIED-QUALITY (KGQQ) THEOREM

This appendix provides the formal proof for the **Knowledge Goal Quantified-Quality (KGQQ)** Theorem presented in Section 3.3. The theorem establishes a theoretical bound on the estimation error of the Knowledge Fitness Criterion (KFC), rigorously linking the stability of the evaluation to the local density of the learned knowledge manifold.

### A.1.1 A.1 PROBLEM SETUP AND DEFINITIONS

Let $(\Omega_K, \mathcal{F}_K, \mu_K)$ be the probability space of knowledge states. Let $(X, \mathcal{F}_X)$ be the measurable feature space (embedding space), where $X \subseteq \mathbb{R}^d$. We define the following measurable mappings and functions:

1. **Knowledge Mapping:** $K : \Omega_K \to X$ maps knowledge states to feature vectors.

2. **Indicator Function:** $I : X \to \mathbb{R}$ represents the ground-truth task metric function defined on the feature space. In practice, $I(x)$ is estimated by a regression model $g_\phi(x)$ based on local samples.

3. **Calibration Function:** $S : \mathbb{R} \to [0, 1]$ maps raw indicators to a standardized score.

**Assumption 1 (Lipschitz Continuity of Calibration):** The calibration function $S$ is Lipschitz continuous with constant $L > 0$. That is, for any $t_1, t_2 \in \mathbb{R}$:

$$|S(t_1) - S(t_2)| \leq L|t_1 - t_2| \tag{9}$$

**Assumption 2 (Local Estimation via Finite Samples):** For any query point $x = K(\omega)$, the estimator for the indicator $I(x)$ is derived from a local aggregate of $m$ independent samples $\{y_1, \ldots, y_m\}$ drawn from the conditional distribution of task outcomes in the neighborhood $\mathcal{N}(x)$. The number of effective samples $m$ is proportional to the local density $\rho_{\mu_X}(x)$ of the induced measure $\mu_X$ on the manifold, i.e., $m \propto \rho_{\mu_X}(x)$. Let $I(x) = \psi(y_1, \ldots, y_m)$ be the aggregation function (e.g., mean).

**Assumption 3 (Bounded Difference):** The aggregation function $\psi$ satisfies the bounded difference property. Changing one sample $y_i$ to $y_i'$ changes the estimate by at most $\frac{c}{m}$, where $c$ represents the intrinsic local variation (boundedness) of the task indicator, denoted as $\sigma_I(x)$ in the main text.

### A.1.2 A.2 THEOREM RESTATEMENT

**Theorem (KGQQ).** For a knowledge state $\omega$ mapped to $x = K(\omega)$, let $\rho_{\mu_X}(x)$ be the local density. With probability at least $1 - \delta$, the estimation error of the KFC score is bounded by:

$$|\epsilon(\omega)| \leq \frac{L \cdot \sigma_I(x) \cdot \sqrt{\ln(2/\delta)}}{\sqrt{\rho_{\mu_X}(x)}} \tag{10}$$

This implies that higher feature density $\rho_{\mu_X}(x)$ leads to tighter error bounds and higher evaluation stability.

### A.1.3 A.3 DERIVATION USING MCDIARMID'S INEQUALITY

**Step 1: Error Decomposition** The estimation error of the KFC score, $\epsilon(\omega)$, is defined as the deviation between the estimated score and the expected score:

$$\epsilon(\omega) = S(\hat{I}(x)) - \mathbb{E}[S(\hat{I}(x))] \tag{11}$$

Using the Lipschitz property of $S$ (Assumption 1):

$$|\epsilon(\omega)| = |S(\hat{I}(x)) - S(\mathbb{E}[\hat{I}(x)])| \leq L \cdot |\hat{I}(x) - \mathbb{E}[\hat{I}(x)]| \tag{12}$$

**Step 2: Application of McDiarmid's Inequality** We focus on bounding the deviation of the indicator estimator $|\hat{I}(x) - \mathbb{E}[\hat{I}(x)]|$. Let $\hat{I}(x) = \psi(Y_1, \ldots, Y_m)$ be a function of $m$ independent random variables. According to Assumption 3, substituting the $i$-th sample changes the function value by at most $c_i = \frac{\sigma_I(x)}{m}$.

**McDiarmid's Inequality** states that for any $\epsilon > 0$:

$$\mathbb{P}(|\hat{I}(x) - \mathbb{E}[\hat{I}(x)]| \geq \epsilon) \leq 2 \exp\left(-\frac{2\epsilon^2}{\sum_{i=1}^m c_i^2}\right) \tag{13}$$

Substituting $c_i = \frac{\sigma_I(x)}{m}$:

$$\sum_{i=1}^m c_i^2 = \sum_{i=1}^m \left(\frac{\sigma_I(x)}{m}\right)^2 = m \cdot \frac{\sigma_I(x)^2}{m^2} = \frac{\sigma_I(x)^2}{m} \tag{14}$$

Thus, the bound becomes:

$$\mathbb{P}(|\hat{I}(x) - \mathbb{E}[\hat{I}(x)]| \geq \epsilon) \leq 2 \exp\left(-\frac{2m\epsilon^2}{\sigma_I(x)^2}\right) \tag{15}$$

**Step 3: Establishing the Density Connection** We set the right-hand side probability to $\delta$ and solve for $\epsilon$.

$$\delta = 2 \exp\left(-\frac{2m\epsilon^2}{\sigma_I(x)^2}\right) \implies \ln(\delta/2) = -\frac{2m\epsilon^2}{\sigma_I(x)^2} \tag{16}$$

Rearranging for $\epsilon$:

$$\epsilon = \sigma_I(x) \sqrt{\frac{\ln(2/\delta)}{2m}} \tag{17}$$

According to Assumption 2, the effective sample size is proportional to density: $m \propto \rho_{\mu_X}(x)$. For simplicity, we let $m \approx \rho_{\mu_X}(x)$ (up to a scaling constant absorbed into $\sigma_I$). Substituting this into the equation:

$$|\hat{I}(x) - \mathbb{E}[\hat{I}(x)]| \leq \frac{\sigma_I(x) \cdot C_\delta}{\sqrt{\rho_{\mu_X}(x)}} \tag{18}$$

where $C_\delta = \sqrt{\ln(2/\delta)/2}$ absorbs the confidence terms.

**Step 4: Final Bound Combination** Substituting the result from Step 3 back into Eq. (12):

$$|\epsilon(\omega)| \leq L \cdot \frac{\sigma_I(x) \cdot C_\delta}{\sqrt{\rho_{\mu_X}(x)}} \tag{19}$$

**Conclusion:** The error bound is inversely proportional to the square root of the local density $\sqrt{\rho_{\mu_X}(x)}$. This rigorously proves that in regions where the knowledge manifold has high density (i.e., where the contrastive learning objective $\mathcal{L}_{contrastive}$ clusters samples), the KFC score exhibits lower variance and higher reliability. $\square$

## A.2 IMPLEMENTATION DETAILS AND HYPERPARAMETERS

To ensure the reproducibility of our KFC framework, we provide detailed specifications of the network architectures, training protocols, and baseline configurations used in our experiments. Our code and pre-trained weights will be made publicly available upon acceptance.

### A.2.1 B.1 NETWORK ARCHITECTURES

The KFC framework consists of two lightweight Multi-Layer Perceptron (MLP) modules: the Manifold Encoder ($f_\theta$) and the Task Indicator Regressor ($g_\phi$). The input dimension $D$ is determined by the hidden state size of the base LLM (DeepSeek-RL-8B), which is $D = 4096$.

Table 3: Architecture of Manifold Encoder $f_\theta$

| Layer | Input Dim | Output Dim | Activation | Details |
|---|---|---|---|---|
| Input Layer | 4096 | 512 | LeakyReLU (0.2) | Linear + BN |
| Hidden Layer | 512 | 128 | LeakyReLU (0.2) | Linear + BN + Dropout(0.1) |
| Output Layer | 128 | 32 | None | Linear (L2 Normalized) |

**1. Manifold Encoder ($f_\theta$):** This module maps high-dimensional LLM hidden states to a low-dimensional compact manifold $\mathcal{M} \subset \mathbb{R}^{32}$. We employ a 3-layer bottleneck architecture with Residual connections and Batch Normalization to facilitate gradient flow and training stability.

**Note on Normalization:** The final output vectors are $L_2$-normalized to lie on the hypersphere unit manifold, which is a prerequisite for the stability of the contrastive loss calculation (Eq. 8).

**2. Task Indicator Regressor ($g_\phi$):** This module maps the manifold embeddings to the scalar task suitability score. Since the task metrics are normalized to $[0, 1]$, we use a Sigmoid activation at the final layer.

Table 4: Architecture of Regressor $g_\phi$

| Layer | Input Dim | Output Dim | Activation | Details |
|---|---|---|---|---|
| Hidden Layer | 32 | 16 | ReLU | Linear |
| Output Layer | 16 | 1 | Sigmoid | Linear |

### A.2.2 B.2 TRAINING HYPERPARAMETERS

We utilize a two-phase training strategy. The hyperparameters were selected based on a grid search on a held-out validation set (10% of samples). All experiments were conducted on a server with $2\times$ NVIDIA RTX 3090 GPUs (24GB VRAM each) and PyTorch 2.1.

Table 5: Hyperparameter Settings for KFC Training

| Parameter | Value | Description |
|---|---|---|
| *General Optimization* | | |
| Optimizer | AdamW | Weight decay set to $1 \times 10^{-4}$ |
| Learning Rate (LR) | $1 \times 10^{-3}$ | Cosine annealing scheduler |
| Batch Size | 256 | Ensures sufficient negative samples |
| Total Epochs | 500 | Early stopping with patience=20 |
| *Manifold Learning (Phase 2)* | | |
| Temperature ($\tau$) | 0.1 | For InfoNCE loss (Eq. 8) |
| Smoothness Weight ($\beta$) | 0.1 | Balances contrastive vs. local structure |
| Feature Dim ($d$) | 32 | Dimension of the target manifold |
| Negative Sampling | In-batch | Randomly sampled from batch ($N-1$ negatives) |
| *Regression (Phase 3)* | | |
| LR (Regressor) | $5 \times 10^{-4}$ | Lower LR for fine-tuning |
| Loss Function | MSE | Mean Squared Error |

### A.2.3 B.3 BASELINE IMPLEMENTATION DETAILS

To ensure a fair comparison, baselines were implemented as follows:

- **Supervised Learning (SL):** An end-to-end MLP with the same architecture as $g_\phi \circ f_\theta$ (i.e., $4096 \rightarrow \cdots \rightarrow 1$), but trained directly on the labeled subset ($N = 490$) using MSE loss without the auxiliary manifold contrastive objective.

- **Rule-Based (RB):** Domain-specific heuristics.
  - *ADS:* Scores calculated based on inverse distance to the nearest obstacle (threshold $< 2m$).
  - *CAMEL:* Keyword density matching (e.g., counting "agreement" or "conflict" terms).
  - *ChatDev:* Static code analysis (syntax error count).

